# The Shifting Depictions of Xiàng in German Translations of the *Dao De Jing*: An Analysis from the Perspective of Conceptual Metaphor Field Theory

**Yubo Zhu [1,2,\*] and Weihan Song [3]**

1   School of German Studies, Shanghai International Studies University, Shanghai 200433, China
2   The Third Department, Luoyang Campus, Information Engineering University, Luoyang 471003, China
3   Independent Researcher, Luoyang 471003, China
\*   Correspondence: zhuyubo@shisu.edu.cn

**Abstract:** Like the Bible, the *Dào Dé Jīng* is one of the most translated classics with worldwide influence, and its translation sets a good example in cross-cultural communication. Among the *Dào Dé Jīng*'s translations, the number of German versions is second only to the English ones. Since its introduction to the German regions, the *Dào Dé Jīng* has been popular among German-speaking scholars and readers, casting profound and far-reaching influences in various fields. Based on the theory of the conceptual metaphor field, the article explores the relationship between *Dào* 道 (way or Dao) and *Xiàng* 象 (Symbolic Imagery, images) in the *Dào Dé Jīng* and builds the mapping from *Xiàng* to *Dào*. In the *Dào Dé Jīng*, Laozi uses images (Xiàng 象) as collective concepts to illustrate his *Dào* and make his idea better understood. Thus, this article focuses on the translation of different key images (Xiàng) in six representative German translations of the *Dào Dé Jīng* and summarizes three main translation techniques used in translating *Xiàng*: shifting, conversion, and concealment. After balancing the cultural differences and translation requirements, the German translators take these techniques to translate *Xiàng* and make relevant concepts more understandable and acceptable for German readers, which facilitates the spread of the *Dào Dé Jīng* in the German regions. Inspired by the German translation of *Xiàng*, contemporary translators shall balance the cultural differences between the source language and target language, choose the appropriate translation strategies and techniques in translating ancient Chinese classics and make their translation a bridge between different civilizations.

**Keywords:** the *Dào Dé Jīng*; *Dào*; conceptual metaphor field; *Xiàng*; translation

## 1. Introduction

As one of China's famous philosophical classics and the sacred book of Daoism, the *Dào Dé Jīng* crystallizes the ancient wisdom and civilization of China. For centuries, the translation of the *Dào Dé Jīng* has attracted the attention of scholars at home and abroad and helped the world know China better. According to Daoxuan's *Collection of Critical Evaluations of Buddhism and Daoism from the Past and Present* (*jí gǔ jīn fó dào lùn héng* 集古今佛道论衡), the *Dào Dé Jīng* was first translated into Sanskrit by Xuanzang and Taoist priests under the official organization in the twenty-first year of Zhenguan in the Tang Dynasty (AD 647) (Liu 2018, p. 234). Later the *Dào Dé Jīng* was translated into Latin, French, German, Japanese, English, and other languages. According to Prof. Misha Tadd, the number of translated versions of the *Dào Dé Jīng* has reached 2000, involving 94 languages (Tadd 2022, p. 88). Second only to the Bible, the *Dào Dé Jīng* is the most translated classic with worldwide influence.

In 1823, French sinologist Jean-Pierre Abel-Rémusat published his *Mémoire sur la vie et les opinions de Lao-Tseu (Memory of the Life and Opinions of Laozi)*, in which he translated five chapters of *Dào Dé Jīng* and elaborated the concept of *Dào*. In 1827, based on French

sinologist Abel Rémusat's French translation, the German philosopher Carl Jos. Hieron Windischmann translated five chapters of the *Dào Dé Jīng* into German[1]. Since then, the translation of the *Dào Dé Jīng* in the German-speaking regions began. The past two centuries witnessed the increase of German translations of the *Dào Dé Jīng* from none to over 150. The number of German versions of the *Dào Dé Jīng* is second only to that of the English ones. Among these German versions of the *Dào Dé Jīng*, excellent translations by Victor von Strauss, Richard Wilhelm, Günther Debon, and so on are widely recognized and quite influential. Meanwhile, the German philosophical circle always pays close attention to Laozi and his work. From Immanuel Kant's strong criticism of Laozi's thinking to the critical acceptance of Georg Wilhelm Friedrich Hegel, Friedrich Wilhelm Nietzsche, Friedrich Wilhelm Joseph Schelling, Martin Buber, and so on to Laozi's ideas, to Martin Heidegger and Karl Theodor Jaspers' appreciation and recommendation of Laozi, Laozi and his *Dào Dé Jīng* gradually came to the fore in the German philosophical circle ([Elberfeld 2000](#)). Likewise, the *Dào Dé Jīng* inspired many writers in German-speaking regions. Laozi's thinking and ideas can be seen in Alfred Döblin's 1915 *Die drei Sprünge des Wang-lun* (*The Three Leaps of Wang Lun*) which sets the story in late eighteenth-century China, Bertolt Brecht's poetry 1924 *Morgendliche Rede an den Baum "Griehn"* (*Morning address to a tree named "Green"*) and 1953 *Eisen* (*Iron*) and his dramas 1941 *Mutter Courage und ihre Kinder* (*Mother Courage and Her Children*), 1943 *Der gute Mensch von Sezuan* (*The Good Person of Szechwan*), Hermann Hesse's 1919 *Demian: Die Geschichte von Emil Sinclairs Jugend* (*Demian: The Story of a Youth*), 1922 *Siddharta: Eine indische Dichtung* (*Siddhartha: An Indian Tale*), and 1932 *Die Morgenlandfahrt: Eine Erzählung* (*The Journey to the East*). Since the 1990s, the promotion of the *Dào Dé Jīng* to the world accelerates with more translations, diversified media, as well as more readers and audiences. The *Dào Dé Jīng* and the philosophical thinking in the book become popular among modern readers, manifesting their vitality in the contemporary world.

Though the *Dào Dé Jīng* is only a book of almost 5000 Chinese letters, it is a book of the world. The *Dào Dé Jīng* covers many fields, such as self-cultivation, state governance, military strategy and tactics, epistemology, cosmology, world and natural outlook, and so on. Meanwhile, the *Dào Dé Jīng* is obscure and hard to read, even for native readers. Without annotations, even Chinese readers cannot interpret the book correctly. However, why did the *Dào Dé Jīng* maintain high popularity in the German world for over a century? This article aims to find the answer from the analysis of the translation of *Xiàng* (Symbolic Imagery)[2] in the German versions of the *Dào Dé Jīng* from the perspective of the conceptual metaphor field. Meanwhile, this article attempts to summarize the techniques used in the translation of the *Dào Dé Jīng* and lay the foundation for the translation of other Chinese classics which bridge the communication between Chinese civilization and other civilization.

## 2. *Xiàng* as the Source Domain and the Conceptual Metaphor Field of *Dào*

Laozi said "The way that can be spoken of is not the constant way"[3] ([Laozi 1963](#), D. C. Lau, trans., p. 5). From the perspective of semantics, *Dào* is an extremely abstract concept that cannot be explained in words. From Wang Bi's explanation that "Semantically speaking, *Xiàng* shares the closest meaning with *Dao*" (*jìn yì mò ruò xiàng* 尽意莫若象), people could find that Laozi sought to make an analogy between *Xiàng* and *Dào* and make himself better understood ([Lou 2011](#), p. 414). Though *Dào* is an abstract concept "that could not be seen, heard, felt, smelled or sensed," still *Dào* could be "embodied by all things or found in all things", and people could feel and understand Dao through the changes of things and their observation and experience ([Rao 2006](#), p. 11). Thus, *Xiàng* could serve as the medium for us to better understand *Dào*.

The original semantic meaning of *Xiàng* is the mammal elephant, but its meaning gradually evolves. In Chapter XXI of *Hán Fēi Zǐ, Illustrations of Laozi's Teaching* (*Hán Fēi Zǐ jiě lǎo* 韩非子解老), the semantic evolution of *Xiàng* is recorded as follows:

> People can rarely see the living elephants, usually the bones of dead elephants, then some people try to represent the living elephants through pictures. With the help of those images, those who have not seen elephants can know what living elephants look like. Therefore, the images (Xiàng 象) are gradually referred to as people's concepts of something.[4] (Zhang 2016, p. 220)

In the Paleolithic Era (or Old Stone Age), China's elephants mainly inhabited the northern regions such as Henan, Shanxi, and Shaanxi. Later, due to the colder climate and the elephants migrated southward. By the Period of Warring States, people in northern China could only see the dead bones of elephants instead of living elephants. Based on the bones, people imagined what living elephants look like (Wang 2013, p. 22). Since then, *Xiàng* has been referred to as what is in people's minds or people's concepts of things. For example, the Chinese letters originate from the inscriptions on bones or tortoise shells of the Shang Dynasty. These inscriptions symbolize the images in the external world. China's ancient calendar was set through the observation of cosmos images. In Chinese medicine, *Xiàng* means manifestation. Viscera are hidden inside the body. As manifestation reflects the condition of viscera and can be observed externally, it is called visceral manifestation (*Zàng Xiàng* 藏象). Likewise, *Xiàng* is an important concept in traditional Chinese aesthetics and painting. Images are widely used in literature and painting to reflect people's feelings. Besides, *Xiàng* is a key term in ancient Chinese philosophy. According to *Zuo Tradition* (*zuǒ zhuàn* 左传), in the third year of Lord Xuan's reign (606 BCE), "the cauldrons were cast with images of various creatures. The hundred things were therewith completely set forth, and the people thus knew the spirits and the evil things" (Zuo 2016, p. 601). Through the casting of images on the cauldrons, "the ancient Chinese developed the early understanding of the relationship between objects and images" (Zhang 2014, p. 69). Likewise, in the *Zhou Book of Change* (*zhōu yì* 周易), *Xiàng* is paraphrased as follows:

> When the sages discovered the esoteric principles under heaven, they compared them to concrete states and appearances, symbolized them with appropriate objects and meaning, and thus called them images. (Ji 2008, p. 383)

With the help of images, the sages make profound and obscure knowledge easier to understand for ordinary people. During the process, *Xiàng* serves as the medium between the physical world and the metaphysical world. In the *Dào Dé Jīng*, Laozi also uses the concrete *Xiàng* to interpret the abstract *Dào*. Such interpretation is based on the common characteristics shared by *Wù* 物(thing), *Xiàng* and *Dào* and their mutual relations. As *Dào* exists in all things (*Wù* 物) and things can be referred to as *Xiàng*, *Xiàng* shares similar, if not all, characteristics with *Dào*. From various perspectives, *Xiàng* is used to interpret *Dào*, to make the intangible tangible, and to make the abstract concrete. Thus, the essence and core of *Dào* are better explained.

To some extent, the relationship between *Xiàng* and *Dào* can also be elaborated by the conceptual metaphor theory. The western conceptual metaphor theory originated from Aristotle's Poetics and Rhetoric, in which he discussed the rhetorical functions of metaphor. In the 1970s, there occurred a cognitive turn in the conceptual metaphor theory. Developed by Lakoff, Johnson, Turner, and other writers, the theory of conceptual metaphor became more and more mature in their books and articles, including *Women, Fire and Dangerous Things: What Categories Reveal about the Mind* (Lakoff 1987), *Metaphors We Live By* (Lakoff and Johnson 1980), *More Than Cool Reason: A Field Guide to Poetic Metaphor* (Lakoff and Turner 1989) (Wang 2007, p. 34). The conceptual metaphor theory holds that metaphors are everywhere in daily life and they serve as conceptual norms which influence human thinking. Based on their experience of the objective world, people can understand the target concepts with relatively weak structures through those with relatively strong structures (Lan 2005, p. 122). Lakoff points out that each metaphor contains a source domain, a target domain, and a source-to-target domain mapping (Lakoff 1987, p. 68); the direction of metaphorical mapping is from concrete to abstract domains (Lakoff 1987, pp. 275–76). Lakoff and Mark Turner further define metaphoric mapping as a set of correspondences between two conceptual domains (Lakoff and Turner 1989, p. 4). However, there can be

more than one source domain or one target domain in conceptual metaphors; in other words, metaphoric mapping does not always occur between two conceptual domains. With examples like "An argument is a container" "An argument is a journey" and "An argument is a building", Kövecses answers "why does a target domain have several origin domains" (Kövecses 2002, pp. 63–64). Kövecses argues that due to the partiality of metaphorical mappings, people tend to specify one target domain with multiple source domains instead of one. Meanwhile, the mapping from source to target domains can also be partial. In the partial metaphorical utilization, only part of the source domain is used in each metaphor. The used part of the source domain is highlighted in forming the target concept, while the part of the target domain that is not highlighted is hidden. The highlighting process is defined as metaphorical highlighting. One source domain can only play a partial role in forming the target domain, but the mystery of the target domain can be revealed with enough source domains. People can get the whole picture of the target domain with a comprehensive understanding of multiple source domains. Together, these source domains build the structure and content of the target domains, facilitating people's understanding of abstract concepts (Kövecses 2002, pp. 84–91).

Kövecses' model of conceptual metaphor focuses on the relationship between the source domain and target domain, while it pays little attention to the relations between different source domains. Inspired by Kövecses' theory, this article attempts to analyze the relationship between *Dào* and *Xiàng* through the mappings from multiple source domains to one target domain. Though the many-to-one mapping model does not reveal the interrelations among different source domains, or *Xiàng* in this article, still Kövecses' theory could bring people closer to the concept of *Dào*.

As shown in Figure 1 and Table 1, [Dào] is the core concept surrounded by the *Xiàng* concepts like *Pǔ*, *Yī*, *Gēn*, *Shuǐ*, *Mén*, *Mǔ*, *Yīng'ér*, *Gǔ* and so on[5]. In the *Dào Dé Jīng*, Laozi uses concepts like *Mǔ* to illustrate his idea of *Dào*, and concepts like *Mǔ* serve as the *Xiàng* (source domain) in building the structure and content of *Dào* (target domain). What lie between [Dào] and *Xiàng* are the semantic features shared by them. In the figure, the two-way arrows are used to reflect the mutual relations between the two ends[6]. For example, the two-way arrows are used to reflect that *Xiàng* shares the semantic features and these concepts share intertextuality. As shown in the figure, the author holds that the interrelated mappings from *Xiàng* and the semantic features to [Dào] form the semantic field of [Dào], serving as the structure and content of [Dào], and bringing people closer to the concept of [Dào].

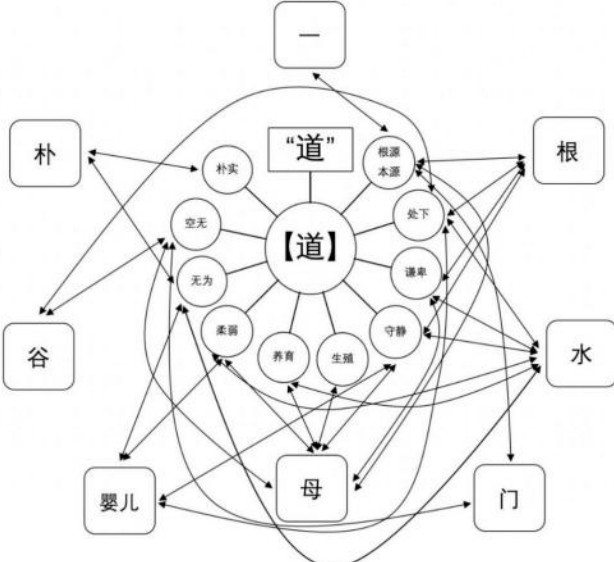

**Figure 1.** The conceptual metaphor field of *Dào* in the *Dào Dé Jīng*.

**Table 1.** Terms in Figure 1.

| Chinese | Chinese Pinyin | English |
|---------|----------------|---------|
| 道 | Dào | Dao |
| 一 | Yī | One |
| 根 | Gēn | Root |
| 水 | Shuǐ | Water |
| 门 | Mén | Gate |
| 母 | Mǔ | Mother |
| 婴儿 | Yīng'ér | Babe |
| 谷 | Gǔ | Valley |
| 朴 | Pǔ | Unpretentious |
| 根源 | Gēn Yuán | Origin |
| 处下 | Chǔ Xià | Below |
| 谦卑 | Qiān Bēi | Humble |
| 守静 | Shǒu Jìng | Stillness |
| 生殖 | Shēng Zhí | Breed |
| 养育 | Yǎng Yù | Nourish |
| 柔弱 | Róu Ruò | Soft and weak |
| 无为 | Wú Wéi | Without doing anything |
| 空无 | Kōng Wú | Nothingness |
| 朴实 | Pǔ Shí | Unadorned |

## 3. The Techniques Used in Translating *Xiàng* in German Translations

The article focuses on the German translations of the *Dào Dé Jīng* and aims to analyze them from the perspective of the conceptual metaphor field of *Dào*. It has been made clear that *Xiàng* acts as the source domain in the conceptual metaphor field of *Dào*. Meanwhile, the mapping from the source domain to the target domain diversifies the content and meaning of the target domain. The mapping process coincides with the translation process which happens from the source language to the target language. Excellent German translations of the *Dào Dé Jīng* combine the mapping process with the translating process to better illustrate the major concepts in the book. Therefore, based on the six representative German translations published in different times, including Victor von Strauss (1870), Richard Wilhelm (2010), Erwin Rousselle (1985), Günther Debon (2014), Viktor Kalinke (2000), and Annette Oelkers (2014), the article attempts to analyze the three main techniques used in translating *Xiàng* in German translations: the shifting of *Xiàng*, the conversion of *Xiàng*, and the concealment of *Xiàng*.

### 3.1. The Shifting of Xiàng

Based on the corpus-based comparative studies on the above six German translations of the *Dào Dé Jīng*, the author finds that the shifting of *Xiàng* is the most commonly used translation technique. In Chinese, *Shifting* is originally a mathematical concept that means moving all points of an object in the same direction by the same distance on the same plane. Shifting doesn't change the shape or size of an object but its position. The author borrows this mathematical term to describe the first technique used in translating *Xiàng* in the German *Dào Dé Jīng*. With such a translation technique, the translated *Xiàng* meets the following conditions: (1) The expressions in the German translation are commonly-used ones that match semantic meanings in Chinese, and the translation does not change the original *Xiàng* and its basic semantics; (2) *Xiàng* simply gets shifted from the original Chinese text to the German one with its relative function in the target text remains.

For example, the [Mǔ] concept appears in the source text in the forms of *Mǔ* 母 (mother), *Cí* 雌 (female), and *Pìn* 牝 (female). The corresponding expressions of this *Xiàng* after being transferred to the German translations are shown in the following table (Table 2):

**Table 2.** Corresponding expressions of [Mǔ] in the German translations of the *Dào Dé Jīng*.

| Translator | *Mǔ* 母 (Mother) | *Cí* 雌 (Female) | *Pìn* 牝 (Female) |
|---|---|---|---|
| Strauss | **Mutter** (mother) | Vogelweibchen (female bird)/**Weibheit** (femininity) | **Weibliche/Weib** (femininity) |
| Wilhelm | **Mutter** | Henne (hen)/**Weibheit** (femininity) | **Weib/Weibliche** (femininity) |
| Rousselle | **Mutter** | Vogelmutter (mother bird)/**Weibheit** (femininity) | Tiergöttin (goddess of animal)/**Weib/weibliches Wesen** (female creatures) |
| Debon | **Mutter** | **Weibchen/Weibheit** (femininity) | **Weibheit/Weib/Weiblichkeit/Weibliche** (femininity) |
| Kalinke | **Mutter** | **Weiblich/Weibliche** (femininity) | **weiblich, Weibliche** (female/femininity) |
| Oelkers | Ursprung/Dao/Mutter | neues Leben entsteht (give birth to new life)/die Fürsorglichkeit, die dem weiblichen Prinzip zugrunde liegt (maternal love) | **Weiblich** (femininity)/ohne Ende wird neues Leben geboren (new life was born inendless circle)/**Weibliche** (femininity) |

As shown in Table 2, the *Xiàng* of [Mǔ] is translated into different German expressions in the six German translations, with those highlighted in bold basically in accord with the Chinese expressions of the *Xiàng*. For example, *Mutter* means *mother*, *Weibheit/Weibliche* and *Weib/Weibliche* both mean *female*, all of which are corresponding expressions of the *Xiàng* of [Mǔ] in German.

A similar example is the translation of [Shuǐ] 水 (water), whose expressions in the original text include *Shuǐ* 水 (water), *Jiānghǎi* 江海 (river or sea), and *Lù* 露 (dew), which when shifted to the German translations become the following ones (Table 3):

**Table 3.** Expressions of [Shuǐ] in the German Translations of the *Dào Dé Jīng*.

| Translator | *Shuǐ* 水 | *Jiānghǎi* 江海 | *Lù* 露 |
|---|---|---|---|
| Strauss | **Wasser** | **Meer, Strom, Fluss, Ozean** | **Tau** |
| Wilhelm | **Wasser** | **Meer, Strom** | **Tau** |
| Rousselle | **Wasser** | **Meer, Strom, See** | **Tau** |
| Debon | **Wasser** | **Meer, Strom** | **Tau** |
| Kalinke | **Wasser** | **Meer, Strom** | **Nass** |
| Oelkers | **Wasser** | **Meer, Fluss, Ozean** | **Tau** |

As shown in Table 3, similar to the translation of [Mǔ], the expressions of [Shuǐ] in the German translations, such as *Wasser* (water), *Strom* (river), *Meer* (sea), *Fluss* (river), *Tau* (dew) and *Nass* (clear water), all fall under the category of [Shuǐ] in the original text. It is another typical example of the shifting of *Xiàng*. In addition, the translations of [Gǔ] 谷 (valley) as *Tal* (valley), and [Pǔ] 朴 (log) as *Rohholz* (log) and *unbearbeiteter Stoff* (unprocessed timber), etc., are examples of the shifting of *Xiàng*.

The shifting of *Xiàng* keeps the content and form of the original text to the greatest extent. It is a translation with an almost original taste and flavor, which not only conforms better to the translation criterion of faithfulness but also ensures the unity of form and spirit of the translated text. However, from the perspective of the target readers, the *Xiàng* in the source language may not be familiar to the target readers, or the translated *Xiàng* means opposite to its original semantic meaning. Thus, the shifting of *Xiàng* may cause two

consequences: first, the same *Xiàng* cannot be understood by the target readers without detailed annotations; second, the semantic asymmetry between the translated text and the source text could cause misunderstanding. Therefore, the shifting of *Xiàng* is the most appropriate technique to preserve both the form and spirit of *Xiàng* in the target language, as long as the *Xiàng* has slight semantic changes, or it is familiar to the target readers. Moreover, from the perspective of the conceptual metaphor field, this pattern keeps the metaphorical mapping relationship and mapping content of the original text.

*3.2. The Conversion of Xiàng*

Through data analysis, the author finds the second translation technique: the conversion of *Xiàng*. That is, the translator, when translating a *Xiàng*, uses another *Xiàng* in the place of the original one, thus introducing a conversion in translation. By further analyzing this technique, the author divides the conversion of *Xiàng* into the following two groups according to the mechanisms:

3.2.1. The Conceptual Conversion

The translators actively adopt translation strategies to avoid the disadvantages of shifting. When cultural differences pose obstacles to the readers' understanding of the original *Xiàng* while it is being transferred from the original text to the target text, the translator uses another *Xiàng* in the target language culture with the same metaphorical meaning to eliminate or reduce the obstacles to the readers. The most typical example of conversion among the six German translations is Erwin Rousselle's translation of [Mén]门 (door). Different from the other five translations which usually use the shifting of *Xiàng* to translate *Mén* 门 (door) from the original text into *Tor* (door) and *Tür* (door) in German, Rousselle translated the two *Mén* in Chapter 1 and Chapter 6 of the *Dào Dé Jīng*, both into *Schoß*, as shown in the following table (Table 4):

**Table 4.** Two translations of [Mén] in Erwin Rousselle's translation.

| Translator | German Translation | Chinese | English Translation |
|---|---|---|---|
| Rousselle | Das Mysterium der Mysterien, aller Geheimnisse **Schoß**. | 玄之又玄，众妙之**门**。——一章 | Mystery upon mystery—The gateway of the manifold secrets. |
| | Der dunklen Tiergöttin **Schoß**, ist Himmels und der Erde Wurzel. | 玄牝之**门**，是谓天地根。——六章 | The gateway of the mysterious female is called the root of heaven and earth. |

As shown in Table 4, the basic semantic meaning of [Mén] in the German cultural context is the passage of entrance and exit, while that of *Schoß* in German is the "pregnant woman's abdomen" or "woman's private parts", which is closely related to productivity. If translated into *Tor* or *Tür*, German readers would not be able to understand [Mén] without detailed annotations. Beyond the *Dào Dé Jīng*, *Schoß* is more relevant to [Mǔ] than to [Mén] both in Chinese and German. The converted *Xiàng* not only affiliates itself with another *Xiàng* ([Mǔ]) in the conceptual metaphor field of *Dào*, but also relates itself to the creativity and originality of *Dào* in the German culture. This cannot be achieved by the shifting of *Xiàng* like *Tor* or *Tür*. The conversion of *Xiàng* not only keeps the expressions about *Dào* as interpreted by *Xiàng* in the original text but also conveys the original and creative semantic features of *Dào*, making it easier for German readers to understand.

3.2.2. The Deviated Conversion

The reason for this conversion is the translator's misunderstanding of the semantics of the original text or the existence of discrepancies between the reference text and the authoritative edition. For example (Table 5):

**Table 5.** The Conversion of *Xiàng* in German Translations.

| Translator | German Translation | Chinese | English Translation |
|---|---|---|---|
| Rousselle | Die Gottheit des Quelltals ist todlos, das ist die dunkle **Tiergöttin**. Der dunklen **Tiergöttin** Schoß ist Himmels und der Erde Wurzel. | 谷神不死，是谓玄**牝**。玄**牝**之门，是谓天地根。——六章 | The spirit of the valley never dies. This is called the mysterious female. The gateway of the mysterious female is called the root of heaven and earth. |
| | Sie ist die tiefe Wurzel und der feste **Stamm**. Die Führerin zu ewigem Leben und dauernder Schau. | 是谓深根固**柢**，长生久视之道。——五十九章 | This is called the way of deep roots and firm stems by which one lives to see many days. |
| Strauss | Nimmt ers leicht, so verliert er **die Vasallen**; ist er unruhig, so verliert er die Herrschaft. | 轻则失**根**，躁则失君。——二十六章 | If light, then the root is lost; If restless, then the lord is lost. |
| Debon | Dieses nennt man: Die Wurzel vertiefen und den **Stamm** festigen. Das ist der Weg ewigen Lebens und dauernder Schau. | 是谓深根固**柢**，长生久视之道。——五十九章 | This is called the way of deep roots and firm stems by which one lives to see many days. |

As shown in Table 5, Rousselle translated the two *Pìn* in Chapter 6 of the *Dào Dé Jīng* into *Tiergöttin* (Goddess of Animals). The title of Rousselle's version of the *Dào Dé Jīng* is called *Lao-tse. Führung und Kraft aus der Ewigkeit* (*Lao-tse. Guidance and Strength from Eternity* 1985); and *Dào* is translated as *Führerin* throughout his translation. Therefore, from the perspective of textual semantics, Rousselle's conversion of *Xiàng* stems from his goddess-based interpretation of [Dào], which is an adaptation in the context of goddess discourse. Strauss converted *Gēn* 根 (root) in Chapter 26 into *Chén* 臣 (minister), a character with irrelevant semantic meanings because he took a different reference from the He Shang Gong Version. (Wang 1993, p. 107) This conversion of *Xiàng* was a de facto conversion, though not a product of Strauss' subjective action. Moreover, Rousselle and Debon both translated *Dǐ* 柢 (root) in Chapter 59 into *Stamm* (tree trunk). Although the trunk and roots are both components of trees, they are not identical parts. More importantly, the semantic features conveyed by them are not the same. The reason for this deviation is probably that Rousselle and Debon had applied the expressions (*Wurzel* and *Stamm*) with a high level of co-occurrence.

From the perspective of conceptual metaphor, although conversion happened after *Xiàng* is translated, the metaphorical mapping relationship still exists because of the adoption of a new *Xiàng* in the translated text, whereas the translation uses a new source domain to describe the original target domain.

### 3.3. The Concealment of Xiàng

Compared with the above two techniques, the concealment of *Xiàng* makes the biggest change during the translation. This means the translator will conceal the *Xiàng* in the source text by completely not using it or only partially using it in the target text. Therefore, according to its concealment extent, the concealment of *Xiàng* in the translation can be divided into two types: the total concealment of *Xiàng* and the partial concealment of *Xiàng*.

### 3.3.1. The Total Concealment of *Xiàng*

This is quite typical in Annette Oelkers' translation, as shown in the following table (Table 6):

**Table 6.** The Total Concealment of *Xiàng* in the German Translation of the *Dào Dé Jīng*.

| Translator | German Translation | Chinese | English Translation |
|---|---|---|---|
| Oelkers | Das Namenlose, das Eine, was wir nicht benennen können, bildet den Anfang von Himmel und Erde. Das mit Namen benannte ist der **Ursprung** der zehntausend Dinge. | 无，名天地之始，有，名万物之**母**。——一章 | The nameless was the beginning of heaven and earth; The named was the mother of the myriad creatures. |
| | Wenn wir uns auf den **Ursprung** besinnen und dies daraus entstandene Welt verstehen, dann können Schwierigkeiten und nichts mehr anhaben. | 既知其子，复守其**母**，没身不殆。——五十二章 | After you have known the child, go back to holding fast to the mother, and to the end of your days you will not meet with danger. |
| | Hat man den **Ursprung** verstanden, dann kann man lange währen. | 有国之**母**，可以长久。——五十九章 | When he possesses the mother of a state, he can then endure. |
| | Der Ursprung des Lebens funktioniert nach dem weiblichen Prinzip; ohne Ende wird neues Leben geboren. Auch der einzelne trägt diese Energie in sich. Der **Ursprung** von Himmel und Erde ist unergründlich. | 玄牝之门，是谓天地**根**。——六章 | The gateway of the mysterious female is called the root of heaven and earth. |
| | Alles erblüht, wieder und wieder, nur aus dem Grund, um zu dem **Ursprung** zurückzukehren; zu dem was ewig ist und ewig sein wird. Diesen ewigen Kreislauf nicht zu erkennen, macht unglücklich. | 夫物芸芸，各复归其**根**。——十六章 | The teeming creatures all return to their separate roots. |
| | In diesem Punkt unterscheide ich mich von anderen Menschen; ich habe erkannt, dass das **DAO** immer für mich sorgen wird. | 我独异于人，而贵食**母**。——二十章 | I alone am different from others and value being fed by the mother. |
| | Kannst du Zugang zum Tor des Lebens haben, ohne dass **neues Leben entsteht**? | 天门开阖，能为**雌**乎？——十章 | When the gates of heaven open and shut, are you capable of keeping to the role of the female? |
| | Der Ursprung des Lebens funktioniert nach **dem weiblichen Prinzip; ohne Ende wird neues Leben geboren**. Auch der einzelne trägt diese Energie in sich. Der Ursprung von Himmel und Erde ist unergründlich. | 玄**牝**之门，是谓天地根。——六章 | The gateway of the mysterious female is called the root of heaven and earth. |
| | Das Schwere schafft die **Grundlage** für Leichtigkeit. Die Ruhe ist das Oberhaupt der Unruhe. . . . Begibst du dich nicht auf die Suche, dann verlierst du **die Verbindung mit dir**. Unruhig zu sein bedeutet, die Herrschaft über die eigenen Gedanken zu verlieren. | 重为轻根，静为躁君。. . . . . . 轻则失**根**，躁则失君。——二十六章 | The heavy is the root of the light; The still is the lord of the restless. . . . If light, then the root is lost; If restless, then the lord is lost. |

As shown in Table 6, in the *Dào Dé Jīng*, she translated *Mǔ* 母 in Chapters 1, 52, and 59 and *Gēn* 根 in Chapters 6 and 16 into *Ursprung* (a German word meaning *origin*). Yet *Mǔ* in Chapter 20 was translated into *Dào*, *Cí* 雌 in Chapter 10 into *Neues leben entsteht* (a German phrase meaning *creating new life*), *Pìn* 牝 in Chapter 6 into *dem weiblichen Prinzip; ohne Ende wird neues Leben geboren* (German phrases meaning *the law of the female; endless creation of new life*), and *Gēn* 根 in Chapter 26 into *Grundlage* (a German word which means *foundation*) or *die Verbindung mit dir* (a German phrase meaning *contact with you*), etc. The reason why Oelkers largely adopted total concealment is that she intended to interpret the

*Dào Dé Jīng* in a chicken-soup style in her translation with more use of close-to-life wording and Free Translation. Therefore, she preferred erasing the *Xiàng* that indirectly indicates the characteristics of *Dào* and directly depicting them in her language. Such translation surely facilitated readers to understand her interpretation of *Dào*, but it also lost the literary and aesthetic value of the source text in which *Xiàng* was used to explain *Dào*.

In addition to the above examples in Annette Oelkers' translation, [Pǔ] was the most totally-concealed *Xiàng* and the only one that was concealed in all six versions of the translation. *Pǔ* 朴 appeared eight times in the source text. As listed in the following table, only the first one was translated with the technique of shifting, the other seven were translated into *Einfalt / Einfältigkeit / einfältig* (German words meaning *simplicity*), *Einfachheit / einfach* (German words that also represent *simplicity*), or *Lauterkeit* (a German word that means *purity*) that can directly indicate their semantic characteristics.

As shown in Table 7, the literal German translation *Pǔ* 朴 is *Rohholz*, the semantic meaning of which is *log*. In German, *Rohholz* has no symbolic meanings, let alone the metaphorical meanings similar to those contained in the source text. Simply shifting *Pǔ* into *Rohholz* would make it hard for German readers to understand the source text. To solve this problem, the translators concealed *Pǔ* and directly presented its metaphorical meanings.

**Table 7.** *Pǔ* 朴 in the German Translation of the *Dào Dé Jīng*.

| Source Language | | Target Language | | | | | |
|---|---|---|---|---|---|---|---|
| | | Strauss | Wilhelm | Rousselle | Debon | Kalinke | Oelkers |
| [Pǔ]朴 | 朴$_1$ | Rohholz | unbearbeiteter Stoff | Rohholz | Grobholz | Holz, das noch nicht beschnitzt ist. | unbearbeitetes Holz |
| | 朴$_2$ | Einfalt | Lauterkeit | Rohholz | das Schlichte | Ursprünglichen | das eigene wahre Wesen |
| | 朴$_3$ | Einfalt | Einfalt | Rohholzsein | Groben und Schlichten | Ursprünglichkeit | —— |
| | 朴$_4$ | Einfalt | Einfalt | Rohholz | Grobholz | Ursprüngliches | die ursprünglichen Eigenschaften |
| | 朴$_5$ | Einfältigkeit | Einfalt | Rohholzhaft | Schlichtheit | Einfachheit | Ursprung |
| | 朴$_6$ | Einfachheit | Einfalt | Rohholzsein | Schlichtheit | Ursprüngliche | Verbindung mit DAO |
| | 朴$_7$ | Einfachheit | Einfalt | Rohholzsein | Schlichtheit | Ursprüngliche | einfach |
| | 朴$_8$ | einfach | einfältig | Rohholz | schlicht | das Einfache (gleich unbeschnittenem Holz) | sind sie selbst |

### 3.3.2. The Partial Concealment of *Xiàng*

This means that some of *Xiàng* are translated into other forms. They are not translated as individual concepts but collateral concepts. The meaning of these *Xiàng* can be seen in different expressions, as shown in the following table (Table 8):

**Table 8.** The Partial Concealment of *Xiàng* in the German Translation of the *Dào Dé Jīng*.

| *Xiàng* (Different Forms) | German/English Translation | Source |
|---|---|---|
| [Mǔ] 母 (*Cí* 雌) | die Fürsorglichkeit, die dem **weiblichen** Prinzip zugrunde liegt (母性般的关怀) | Oelkers: Chapter 28 |
| [Gǔ] 谷 (*Gǔ* 谷) | **Thal**niederung (谷之低) | Strauss: Chapter 28 |
| [Gēn] 根 (*Gēn* 根) | **Wurzel**grund (根基) | Debon: Chapter 16, 26 |
| [Gēn] 根 (*Gēn*根、*Dǐ* 柢) | mit dem Ursprung **verwurzelt** (扎根) | Oelkers: Chapter 59 |
| [Pǔ] 朴 (*Pǔ* 朴) | das Einfache (**gleich unbeschnittenem Holz**) (像未雕琢过的木头一样简单) | Kalinke: Chapter 57 |

In the above examples in Table 8, *Xiàng* concepts like [Mǔ] [Gǔ] [Gēn] and [Pǔ] are not translated as individual concepts, but their semantic features are kept in other forms. The words in bold in Table 8 like *weiblich* (female), *Tal* (valley), *Wurzel* (root), and *Holz* (log) maintain the features of the corresponding *Xiàng*. Though these words are attributives or compound words, the semantic features of the *Xiàng* are concealed in them. The partial concealment of *Xiàng* essentially takes the form of the "*Xiàng* + its metaphorical meaning." Together such form shall be regarded as simile, instead of metaphor.

## 4. The Differences between the Transfer Modes of *Xiàng* in German Translations

Based on the above analysis, the author lists the features of the transfer modes of *Xiàng* in German translations in Table 9 to examine the source domain, target domain, and the mapped semantic features (common features shared by *Dào* and *Xiàng*) before and after the translation.

**Table 9.** Features of the Transfer Modes of *Xiàng* in German translations.

| | Shifting of *Xiàng* | Conversion of *Xiàng* | | Concealment of *Xiàng* | |
|---|---|---|---|---|---|
| | | Conceptual Conversion | Deviated Conversion | Total Concealment | Partial Concealment |
| Source Domain | Unchanged | Changed | Changed | Concealed | Unchanged |
| Target Domain | Unchanged | Unchanged | Unchanged | Concealed | Unchanged |
| The Mapped Semantic Features | Unchanged | Unchanged | Changed | The metaphor disappears and the semantic features are expressed explicitly. | The semantic features remain unchanged while the metaphors are changed to simile or fixed expression in German. |

According to Table 9, we can find that: (1) Through the shifting of *Xiàng*, the source domain, target domain, and mapped semantic features remain unchanged after translation. (2) Through the conceptual conversion of *Xiàng*, the target domain and mapped semantic features remain unchanged while the source domain changes. However, if the translation is based on misreading or the different versions of the original text, only the target domain remains unchanged. (3) In the total concealment of *Xiàng*, the conceptual metaphors in the original text are not translated; the source domain and target domain are concealed, while the mapped semantic features are expressed explicitly. In the partial concealment of *Xiàng*, the mapped semantic features in the source language remain unchanged while the conceptual metaphors are changed to similes or fixed expressions in German which are usually ignored.

Judging from the effects of the different translation modes, through the shifting of *Xiàng*, the three elements of conceptual metaphor remain unchanged. However, such a mode does not always mean it is the best translation technique in dealing with Chinese classics. For example, Strauss and Wilhelm take such a mode in translating the *Dào Dé Jīng* in the late 19th and early 20th centuries, but their target readers are usually missionaries, sinologists, philosophers, and other professional scholars who are familiar with Chinese culture. Therefore, they use the shifting of *Xiàng* to maintain "both the form and spirit" of the original texts. Meanwhile, there are many annotations in their versions of the *Dào Dé Jīng*. Given the detailed explanation and annotation in the preface of Strauss's translation, Strauss knows the huge differences between the two cultures and tries to bring German readers closer to the Chinese culture. Next to the shifting of *Xiàng*, the other two modes are also indispensable in translating Chinese classics like the *Dào Dé Jīng*. They all play essential roles in translation and cross-cultural communication.

## 5. Conclusions

In the past two centuries, excellent German translations of the *Dào Dé Jīng* kept on popping up, making the *Dào Dé Jīng* popular in the German regions and influencing people in various fields. Thanks to German translators' flexible translation techniques used in translating *Xiàng* 象 (Symbolic Imagery, image), the gist of the *Dào Dé Jīng* and Laozi's philosophical thinking becomes understandable and acceptable to readers. In the *Dào Dé Jīng*, Laozi uses concrete *Xiàng* to illustrate the abstract *Dào*, and these *Xiàng* concepts serve as the source domains of the target domain, namely *Dào*. Gradually the content and semantic features of *Dào* become diversified and enriched. Meanwhile, compared to *Dào*, *Xiàng* is easier to understand and more accessible in people's daily life. From their daily experience, people gradually have a comprehensive and holistic view of *Dào*. Thus, the translation of *Xiàng* becomes a key issue and a tough issue in the translation of the *Dào Dé Jīng*. Based on the conceptual metaphor theory developed by Lakoff, Johnson, Turner, Kövecses, and other linguists, the author focuses on the translation of *Xiàng* concepts in six representative German versions of the *Dào Dé Jīng* and summarizes three techniques used in translating the *Dào Dé Jīng*: the shifting, conversion, and concealment. These techniques make the abstract *Dào* translatable and bring Laozi's ideas closer to German-speaking readers.

Instead of focusing on the pros and cons of different German versions of the *Dào Dé Jīng*, this article focuses on analyzing the translation techniques or the transfer modes of *Xiàng* in German translations of the *Dào Dé Jīng*. The flexible translation techniques used by the German translators made *Dào Dé Jīng* popular in the German-speaking regions. Through the shifting of *Xiàng*, German translators attempted to find the replacement of the concepts of the *Dào Dé Jīng* in their native language, making small shifts between the source language and target language. Meanwhile, they use detailed annotations to illustrate the abstract concepts, maximizing the illustration of *Dào* to readers. Without these annotations, their translation will be confusing and obscure. Through the conversion of *Xiàng*, German translators narrow the cultural differences between the source and target language. Some of the Chinese *Xiàng* concepts in Chinese are conversed with new German concepts to facilitate German readers' understanding of *Dào*. With the help of the conversed German concepts, the German readers feel close to these unknown Chinese concepts and have a holistic view of the semantic features of *Dào*. Through the concealment of *Xiàng*, German translators stick to the principle that "less is more." German translators choose not to translate some of the *Xiàng* concepts to avoid making the readers confused. Though such concealment does not convey the original linguistic and aesthetic features of the texts, the semantic features and meaning are maintained. It is regrettable to make such concealment, but such concealment can facilitate people's understanding of the text.

To sum up, the flexible translation of *Xiàng* in German versions of the *Dào Dé Jīng* inspires future translations of ancient Chinese classics. The translation of Chinese classics needs more than the translators' proficient language skills in dealing with unique Chinese concepts and terms; it also requires the translators to have profound knowledge of different cultures and sharp conceptions of cultural differences. It is easy to translate the words but not the thoughts. Rigid translation of culture can only convey words not thoughts. Contemporary translators need to pay more attention to the cultural backgrounds of their target readers and choose words carefully, turning their translation into a bridge of cross-cultural communication. Though modern translators have done a great job in their work, they still have a long way to go.

**Author Contributions:** Conceptualization, Y.Z.; methodology, Y.Z.; formal analysis, Y.Z.; resources, Y.Z. and W.S.; data curation, Y.Z.; writing—original draft preparation, Y.Z.; writing—review and editing, Y.Z. and W.S.; project administration, W.S.; funding acquisition, Y.Z. All authors have read and agreed to the published version of the manuscript.

**Funding:** This research was funded by China Postdoctoral Science Foundation: 2021M702208.

**Conflicts of Interest:** The authors declare no conflict of interest.

## Notes

[1] From 1827 to 1834, Carl Jos. Hieron Windischmann published a four-volume monograph *Die Philosophie im Fortgange der Weltgeschichte* (*Philosophy in the Process of World History*) in Bonn, Germany. In the first volume *Die Grundlagen der Philosophie im Morgenlande* (*The Foundation of Eastern Philosophy*), Windischmann translated five chapters (Chapters 1, 14, 25, 41, 42) of *Dào Dé Jīng* from Rémusat's French translation.

[2] *Xiàng* is a unique philosophical concept in Chinese classics, and it shares similar but not equivalent meanings with western philosophical terms such as symbolic imagery and image. To distinguish this Chinese philosophical concept from the western ones and explain it in the context of Chinese culture, this article uses Chinese pinyin *Xiàng* to refer to this concept.

[3] All the English quotes from the *Dào Dé Jīng* are based on D. C. Lau's translation.

[4] The English quotes from *Hán Fēi Zǐ* are translated by the author.

[5] In *Dào Dé Jīng*, the same Chinese character, such as *Mǔ* 母 (mother), could mean both the symbolic imagery and the specific word. To avoid confusion, this article uses "[ ]" to distinguish the symbolic imagery from the words. For example, the [Mǔ] serves as the *Xiàng*, the medium, or source domain in understanding *Dào*, while *Mǔ* just means mother. As in Figure 1, the [Dào]【道】in the core means the core concept of *Dào Dé Jīng*, while the *Dào* "道" above the [Dào] refers to the Chinese letter *Dào* 道.

[6] To highlight the relations between different concepts and make the picture clear, the writer only lists part of the arrows, and those not listed can be inferred from the figure.

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
