# Peer review of "The Shifting Depictions of Xiàng in German Translations of the Dao De Jing: An Analysis from the Perspective of Conceptual Metaphor Field Theory"

_religions, doi:10.3390/rel13090827_

Round 1
Reviewer 1 Report
suggestions:
Richard is the given name, Wilhelm the surname. This is several times confused (p. 6 Richard instead of Wilhelm), same p. 11
Dao de jing is also very important in the New Age movement and esoterics.
p. 1 line 40: German speaking regions
p. 1 line 38: Jean-Pierre Abel-Rémusat wrote a book about Daoism including the translation of some important chapters, not the whole book. Mention the title
p. 2: include given names to the surnames. Formulate the idea a bit clearer: influence of Daoism onto German philosophy and literature.
p. 2 line 96: bones of dead elephants
5 line 203: meets
Author Response
Response to Reviewer 1 Comments
Dear Reviewer:
Thank you for your constructive comments and suggestions on my manuscript. We have carefully considered your suggestions and made some revisions. Over the past several days, we tried our best to polish and revise our manuscript.
The yellow part has been revised according to your comments. Revision notes, point-to-point, are given as follows:
Point 1: Richard is the given name, Wilhelm the surname. This is several times confused (p. 6 Richard instead of Wilhelm), same p. 11
Response 1: Sorry for this confusion, and we have corrected this mistake according to your suggestion.
Point 2: Dao de jing is also very important in the New Age movement and esoterics.
Response 2: We do agree with you that Dao de jing is also very important in the New Age movement and esoterics. The early translations of Dao de jing in Western countries were usually made by middle-age missionaries. These middle-age translations of Dao de jing facilitated the cultural communication between the Western world and China and they brought the Western people different philosophic ideas and worldviews. In the New Age movement, Buddhism and Daoism were very popular in the United States. Dao de jing was quite influential among New Age movement activists, like Gia-Fa Feng.
Point 3: p. 1 line 40: German speaking regions
Response 3: Sorry for this mistake. We have changed the “German speaking regions” into “German-speaking regions”.
Point 4: p. 1 line 38: Jean-Pierre Abel-Rémusat wrote a book about Daoism including the translation of some important chapters, not the whole book. Mention the title
Response 4: In 1832, French sinologist Jean-Pierre Abel-Rémusat published his Mémoire sur la vie et les opinions de Lao-Tseu, in which he translated five chapters of Dào Dé Jīng and elaborated the concept of Dào. In 1827, based on Rémusat’s French translation, the German philosopher Carl Jos. Hieron Windischmann translated five chapters of the Dào Dé Jīng into German.
Point 5: p. 2: include given names to the surnames. Formulate the idea a bit clearer: influence of Daoism onto German philosophy and literature.
Response 5: Sorry for the missing given names. We have included the given names to the surnames in the article. As for the influence of Daoism onto German philosophy and literature, we realize it is a big topic and we might fail to clarify it in this article. Indeed, the influence of Daoism onto German philosophy and literature is a great topic for a book. We are glad to have your suggestion and we will keep on working on that and try to clarify that in our future papers and books.
Point 6: p. 2 line 96: bones of dead elephants
Response 6: Sorry for this mistake. We have revised it according to your suggestion.
Point 7: 5 line 203: meets
Response 7: Sorry for this mistake. We have revised it according to your suggestion.
Abel-Rémusat, Jean-Pierre. 1823. Mémoire sur la vie et les opinions de Lao-Tseu. Paris : Impr. Royale.
Reviewer 2 Report
Notes for The Shifting Depictions of Xiàng
This paper has potential in its title. But it seems to me that the paper is asking for a clear, succinct, and even un-Daoist reading of the book Dào Dé Jīng/The Classic of the Way and Virtue/The Book of the Way and Regulation? Just to translate the title is a task. I am not convinced the paper is systematic in framing semantic elements in Laozi. Yes, you have set up charts and boxes to contain certain linguistic elements, but the methodology of xiang escapes me, especially since the author refuses to acknowledge the possible semantic tone (i.e., the meaning) of the word. Why not go with Hanfei? I think he was able to demystify some of Laozi (I need to read more of 韩非子解老). The passage used at the beginning of the article is truly fascinating:
“People can rarely see the living elephants, usually the bones of the dead ones, then some people try to represent the living elephants through pictures. With the help of those images, those who have not seen elephants can know what living elephants look like. Therefore, the images (Xiàng 象) are gradually referred to as people’s concepts of something” (Zhang Jue 2016, p.220).
The author might as well take credit for the translation. Didn’t she or he translate this text? Surely it was not Zhang Yue? One of the most curious things is the relationship between Daoism and Legalism. And Hanfei’s text is particularly important here. I am not going to translate this passage because it would take too much time for me. But clearly Hanfei is hinting at a philosophy of language here, as he does in many places. Hanfei is the philosopher of power relations. Here the relation is between a word and the dead remains of a thing. So the bones of dead elephants are represented by Xiang? The problem as I see it is not that the word that stands in for the bones, the only things left for people to see since the elephants are dead (All elephants? Is there an elephant shortage?). The problem is not how one word represents a living elephant, but how the words that mimic bones come to represent resemblance. And why is the elephant the animal that becomes the word resemblance? Where did these dead elephants exist in China?
Does the author have a definition for metaphor?
Below are just some responses to the article.
p. 1: “Second only to the Bible, the Dào Dé Jīng is the most 36 translated classic with worldwide influence.”
This sentence contradicts itself. Dào Dé Jīng cannot be second only to The Bible and at the same time the most translated classic (Hm. Sounds like Hanfei’s contradiction).
p. 1: “what is the “the German region”?
Is the German region referring to countries where German is spoken or Germany?
p. 2: 2. “Xiàng as the Source Domain and the Conceptual Metaphor Field of Dào. Xiàng as the Source Domain and the Conceptual Metaphor Field of Dào” Laozi said “The Tao that can be trodden is not the enduring and unchanging Tao” (James Legge 2008, p.8).”
poor Legge, how I miss you. James Legge is an important, extremely important, Westerner (in some ways for the West like Yan Fu for turn-of-the-century China). But “The Tao that can be trodden!!!!! . . . ? Is this a literalist translation problem? Legge published an article about the proper way to translate God into Chinese. But I do not know what it means to “trod.” To take a walk in the moors of Scotland? 道可道非trod 也.
p. 2-3
that list of authors who’ve talked about or referred to Laozi is long, but I am wondering whether a short summary of the respective discourses of Laozi wouldn’t help the appeal to authority this paragraph represents. So Hegel read Laozi, so what. In the 1980s linking Heidegger to Buddhism was quite popular. I did it myself for a well-meaning but shitty paper for my “Introduction to Phenomenology” class. But no amount of name dropping helps an argument unless the author is able to outline, even generally, the names he has chosen to include in his (meant to be impressive) list.
p. 5: “For example, the two-way arrows between the Xiàng concepts and the semantic features reflect that Xiàng shares the semantic features or the semantic features can be seen in the Xiàng.”
This sentence is tautological. If I share can I enter?
p. 6:
This article is in English. You are discussing Chinese-German translation. I think it would help for an English reader if you put the German translation terms in some kind of English. I can read German words and copy and paste them into google translate, and I love Rammstein, but when I read an article I want the author to explain everything. Especially non-English words in English, the language of the journal it may be published in.
p. 6:
so I understand there is a shifting of Xiàng. But what pray is the meaning of the shifting of Xiàng? What is the meaning of the shifting of Xiàng? Presumably this paper was written to describe and outline the meaning of the shifting of Xiàng in The Book of the Way and State Virtue (why not?). Why not flex for Deleuze and Guattari? Take and bite of the post-modern in all of us?
p. 7: “The shifting of Xiàng keeps the content and form of the original text to the greatest extent. It is a translation with an almost original taste and flavor, which not only conforms better to the translation criterion of faithfulness but also ensures the unity of form and spirit of the translated text.”
So far so Xiàng. But I would love to read a definition of Xiàng. Just for reference. To understand the article. I do not feel like I can read Xiàng here and have a clear idea of what you are implying. Is Xiàng a kind of methodology? Or just good translation?
p. 7: “Through data analysis.”
Have we launched into hyper-data here? Has the author counted how many times the word Xiàng appears in 道德经?
p. 8: So 玄牝is Tiergöttin? So (to use the current rhetorical) wtf is Tiergöttin? The word in Chinese is one thing, but looking for an obscure form of a German word is another.
p. 9:
I’m not sure what happened. The author flashes Ursprung, an incredibly complex German word with many resonances and connotations as a complete hiding of Xiang, a word which does not appear in this passage, and which remains defined only in passing.
Author Response
Response to Reviewer 2 Comments
Dear Reviewer:
Thank you for your constructive comments and suggestions on my manuscript. We have carefully considered your suggestions and made some revisions. Over the past several days, we tried our best to polish and revise our manuscript.
The yellow part has been revised according to your comments. Revision notes, point-to-point, are given as follows:
Point 1: p. 1: “Second only to the Bible, the Dào Dé Jīng is the most 36 translated classic with worldwide influence.”This sentence contradicts itself. Dào Dé Jīng cannot be second only to The Bible and at the same time the most translated classic (Hm. Sounds like Hanfei’s contradiction).
Response 1: Sorry for this mistake. We have corrected this mistake according to your suggestion. “Like the Bible, the Dào Dé Jīng is one of the most translated classics with worldwide influence, and its translation sets a good example in cross-cultural communication.”
Point 2: p. 1: “what is the “the German region”? Is the German region referring to countries where German is spoken or Germany?
Response 2: Sorry for this mistake. We have changed the “German speaking regions” into “German-speaking regions”.
Point 3: p. 2: 2. “Xiàng as the Source Domain and the Conceptual Metaphor Field of Dào. Xiàng as the Source Domain and the Conceptual Metaphor Field of Dào” Laozi said “The Tao that can be trodden is not the enduring and unchanging Tao” (James Legge 2008, p.8).”
poor Legge, how I miss you. James Legge is an important, extremely important, Westerner (in some ways for the West like Yan Fu for turn-of-the-century China). But “The Tao that can be trodden!!!!! . . . ? Is this a literalist translation problem? Legge published an article about the proper way to translate God into Chinese. But I do not know what it means to “trod.” To take a walk in the moors of Scotland? 道可道非trod 也.
Response 3: We do agree with you that Legge’s translation is not a good one. His translation is a common literalist translation problem and his understanding of the Dào is a little bit rigid. In Chinese, Dào道(way or Dao) could be a noun, a verb, or both. As a noun, Dào道(way or Dao) means the rules of the world, the order of the world, the faith, the virtue, the way, the road and so on. As a verb, Dào道(way or Dao) means “to tell someone something”, “to walk”, and so on. Then, Laozi’s “道可道,非常道” can be explained in many ways. I prefer that “The rules of the world can be felt, but cannot be told in a few words”. This is my personal understanding of the first sentence of Dào Dé Jīng. Even I know Legge’s translation is problematic, still I quote from him since his translation is one of the most popular ones in the Western world. Besides, you have mentioned Yan Fu, one of the most influential translators in modern history of China. He was important, but not that important, especially after years of translation study progress in China. Both James Legge and Yan Fu were forerunners in the history of translation study, but they are not the best runners. Best runners are those running with the time and the Change. We’d like to see more and better translations of Dào Dé Jīng in the world, if you know some, please do tell us. Thank you in advance.
Point 4: p. 2-3 that list of authors who’ve talked about or referred to Laozi is long, but I am wondering whether a short summary of the respective discourses of Laozi wouldn’t help the appeal to authority this paragraph represents. So Hegel read Laozi, so what. In the 1980s linking Heidegger to Buddhism was quite popular. I did it myself for a well-meaning but shitty paper for my “Introduction to Phenomenology” class. But no amount of name dropping helps an argument unless the author is able to outline, even generally, the names he has chosen to include in his (meant to be impressive) list.
Response 4: We are glad to have your constructive suggestions. Essentially, your question could be summarized into the influence of Daoism on German philosophy and literature, we realize it is a big topic and we might fail to clarify it in this article. Indeed, the influence of Daoism onto German philosophy and literature is a great topic for a book. We are glad to have your suggestion and we will keep on working on that and try to clarify that in our future papers and books.
Point 5: p. 5: “For example, the two-way arrows between the Xiàng concepts and the semantic features reflect that Xiàng shares the semantic features or the semantic features can be seen in the Xiàng.”This sentence is tautological. If I share can I enter?
Response 5: Sorry for the wordy and redundant sentences. As you have mentioned, we use some charts and boxes to contain certain linguistic elements. We want to explain every elements in the charts, but we ignore that some of the elements does not need any explanations since they are easy to understand. Thus, we delete some parts of the sentences to make they more concise. Indeed, brevity is the soul of wit.
Point 6: p. 6: This article is in English. You are discussing Chinese-German translation. I think it would help for an English reader if you put the German translation terms in some kind of English. I can read German words and copy and paste them into google translate, and I love Rammstein, but when I read an article I want the author to explain everything. Especially non-English words in English, the language of the journal it may be published in.
Response 6: Sorry for the missing explanations. Actually, only a few German words are used in the article. We have provided relevant English translations behind the German words or sentences. With the help of Google Translator and dictionaries, the language barriers are few, but something still gets lost in translation. We will try to provide more information about certain words or terms in this article.
Point 7: p. 6: so I understand there is a shifting of Xiàng. But what pray is the meaning of the shifting of Xiàng? What is the meaning of the shifting of Xiàng? Presumably this paper was written to describe and outline the meaning of the shifting of Xiàng in The Book of the Way and State Virtue (why not?). Why not flex for Deleuze and Guattari? Take and bite of the post-modern in all of us?
Response 7: Actually, in Chinese, the “shifting” means “转换” which consists of “转” and ”换”. “转” means “spin”, “rotate”, “converse”, “turn around”, and so on. ”换” means “replace”, “in exchange”, and so on. The earth rotates, so the day and night replaces each other. Then, we come back to the shifting of Xiàng. As you know, Xiàng is difficult to explain or elaborate. Thus, we have to use some simpler concepts to “replace” the Xiàng and make it easier to understand.
Here, we want to say more about Xiàng. The Xiàng used in the title and article is usually the Greatest Xiàng (象, symbolic imagery, images). In chapter 41 of Dào Dé Jīng, “大音希声,大象无形” was mentioned. Our understanding or translation of this sentence is that “The greatest music is silence, and the greatest image has no form”. Modern Chinese letters originates from the inscriptions on bones or tortoise shells. The early Xiàng (象) looks like this which is obviously an abstract painting of an elephant. And original meaning of Xiàng (象) refers to the elephant. You can read relevant part in our article. “People can rarely see the living elephants, usually the bones of dead elephants, then some people try to represent the living elephants through pictures. With the help of those images, those who have not seen elephants can know what living elephants look like. Therefore, the images (Xiàng象) are gradually referred to as people’s concepts of something (Zhang Jue 2016, p.220).”
The evolution of Chinese letter 象 from the inscriptions on bones or tortoise shells to modern Chinese reflects that Xiàng gradually gets rid of its form and it gradually becomes the greatest Xiàng without form. By the way, Zhang Jue translated Han Fei’s Illustrations of Laozi’s Teaching from ancient Chinese to modern Chinese. The ancient Chinese and modern Chinese are almost two languages. The former is much more concise and succinct than the latter. Thus, the ancient Chinese is more difficult to understand. Based on Zhang Jue’s modern translation and Zhu Yubo’s understanding of the ancient Chinese, we translated the quotation from Illustrations of Laozi’s Teaching into English. We have explained the origin of the Xiàng (象) and how the Xiàng (象) are gradually referred to as people’s concepts of something. As the greatest Xiàng has no form, if we want to use some simpler concepts to “replace” the Xiàng and make it easier to understand, we need to use images with specific forms to explain the greatest Xiàng. That is why in this paper, images like river, valley and mother are widely used and discussed. We hope our explanation could mitigate your confusion. The ancient Chinese is difficult enough for our tiny paper, not to mention Deleuze and Guattari. We want to combine the modern with the ancient, but maybe Plato’s Republic, especially the cave theory, is more helpful to the interpretation of Xiàng .
Point 8: p. 7: “The shifting of Xiàng keeps the content and form of the original text to the greatest extent. It is a translation with an almost original taste and flavor, which not only conforms better to the translation criterion of faithfulness but also ensures the unity of form and spirit of the translated text.”
So far so Xiàng. But I would love to read a definition of Xiàng. Just for reference. To understand the article. I do not feel like I can read Xiàng here and have a clear idea of what you are implying. Is Xiàng a kind of methodology? Or just good translation?
Response 8: Sorry for making you confused about Xiàng. We have discussed a lot about Xiàng in the above part. We do hope you have more understanding about after our explanation.
Point 9: p. 7: “Through data analysis.”Have we launched into hyper-data here? Has the author counted how many times the word Xiàng appears in 道德经?
Response 9: We collect over 100 translations of Dào Dé Jīng and create a corpus with these translations. In this paper, we selected six representative translations for further analysis. If you need relevant translations, we could provide them for you.
Point 10: p. 8: So 玄牝is Tiergöttin? So (to use the current rhetorical) wtf is Tiergöttin? The word in Chinese is one thing, but looking for an obscure form of a German word is another.
Response 10: In Chinese, “玄” means “black or deep color”, and “牝” means “doe”, and together “玄牝” means “black doe”. Since the doe and deer are regarded as animals with deity in China, then “玄牝” could be interpreted as some kind of goddess or mystic female. According to the definitions in Cambridge Dictionary, “Tier“ means animal or animal other than man, and “Göttin” means goddess. Then, Tiergöttin could be understood as goddess of animal. It partially explains why “玄牝” was translated into Tiergöttin.
Point 11: p. 9: I’m not sure what happened. The author flashes Ursprung, an incredibly complex German word with many resonances and connotations as a complete hiding of Xiang, a word which does not appear in this passage, and which remains defined only in passing.
Response 11: As we have provided more information about Xiàng, you might find that Ursprung and Xiàng are different concepts. We do appreciate that you points out the resonances and connotations of Ursprung and we will dig deeper into that concept and make comparative study between Ursprung and Xiàng. Yet in this paper we want to focus more on Xiàng.
In all, we appreciate your suggestions and comments. Some of your suggestions are great. However, in this article, we just focus on translation study and linguistics. As you have pointed out, it is difficult enough to just translate the title of Dào Dé Jīng not to mention combining it with modern theories. We still have a long way to go, but we will keep on moving and make our study further and deeper.

Reviewer 3 Report
This is a valuable contribution based on careful translatological analysis of an understudied aspect of Chinese literature and translation studies. It is well-structured and presented, convincingly argued and manages to navigate the various fields it touches on without losing accessibility.
Author Response
Response to Reviewer 3 Comments
Point 1: This is a valuable contribution based on careful translatological analysis of an understudied aspect of Chinese literature and translation studies. It is well-structured and presented, convincingly argued and manages to navigate the various fields it touches on without losing accessibility.
Response 1:
Dear Reviewer:
Thank you for your support and encouragement. We are very glad to have your comments. It thrills us to read that. Still, over the past several days, we tried our best to polish and revise our manuscript. We want to make it better and present our revised manuscript for you.
Thank you in advance for your kind cooperation and we look forward to hearing from you soon.
Kind regards.